# SaMoE: Parameter Efficient MoE Language Models via Self-Adaptive Expert Combination

## Abstract

Recently, Mixture-of-Experts (MoE) has demonstrated success in scaling models to have large amounts of parameters without significant increases in computational cost. However, MoEs have been also reported to be parameter inefficient such that larger models do not always lead to better performance. In this work, we study how to build parameter-efficient MoE models. Our analysis identifies that MoE layers exhibit poor gradient flow as the number of experts increases, leading to insufficient training of experts. To overcome this issue, we propose a new MoE architecture design (SaMoE), which improves the parameter-efficiency of MoE models by learning a soft combination of a global set of expert layers for each MoE layer. Such a scheme enables substantial parameter savings on MoE while achieving comparable or better accuracy than the standard MoE training baseline. Extensive experiments on billion-scale GPT-3 style autoregressive MoE language models demonstrate that SaMoE significantly improves the parameter efficiency of MoE models by reducing up to $5.2\times$ total parameters while obtaining superior pre-training and zero-shot generalization results as compared to baseline.

## 1 Introduction

Over the past few years, there has been an explosion in research revolving around large language models, primarily motivated by the impressive performance of Transformer-based language models (Devlin et al., 2019; Radford et al., 2019; Raffel et al., 2019; Brown et al., 2020). One of the most impactful findings of this research is that the performance of these models continues to scale as the number of parameters increases (Kaplan et al., 2020; Clark et al., 2022). However, sustaining model parameters growth is getting more challenging due to the increasing compute requirements. As such, there has been substantial interest in exploring more efficient model designs and training methodologies. Among them, sparsely activated models, such as architectures based on Mixture-of-Experts (Shazeer et al., 2017; Lepikhin et al., 2020; Fedus et al., 2021), have demonstrated promising results for training massive language models. MoE allows each input to interact with only a subset of the network parameters – chosen independently for each input. As such, the number of parameters is nearly disentangled from the computation cost of processing an input.

Recently, several works explored whether MoE models can be comparatively accurate to dense models but with much lower computational costs. As a result, they have successfully trained MoE-based language models and demonstrated that MoE models could perform on par with their dense equivalent counterparts but with up to $4\text{-}7\times$ reduction in computation cost (Artetxe et al., 2021; Du et al., 2022; Rajbhandari et al., 2022). Despite promising results, MoE architecture appears to be parameter inefficient, considering the yielded model quality improvement vs. the involved parameters. For example, prior works report that to achieve the same quality as the dense model, the MoE model has roughly an order of magnitude more parameters than its corresponding dense model (Rajbhandari et al., 2022; Du et al., 2022; Artetxe et al., 2021). This parameter inefficiency adds a high cost of using additional memory and devices during model training and inference. Therefore, a natural question to ask is: "Are all these expert parameters necessary to increase the model quality?" or equivalently, "*Given a bound on the number of trainable parameters of a model, how can we arrive at an MoE model with higher quality?*"

In this work, we investigate parameter-efficient architectures for MoE. In particular, our analysis shows that MoE models face challenges of poor gradient flow at MoE layers, leading to insuffi-

cient training of those layers compared to the dense layers. Based on this analysis, we conjecture that sharing parameters across experts would allow experts to receive more sufficient training and become useful. As such, we study several expert sharing strategies for MoE models. Our studies show that due to the smaller number of parameters, the performance of MoE models with aggressive tied-experts suffers when training on large-scale GPT pretraining datasets. On the other hand, relaxing the expert sharing constraints helps improve the model quality, but it requires manually designing the sharing strategy and the manually determined strategy may still achieve sub-optimal model quality. Our contributions in this work are:

**SaMoE**. We improve the parameter efficiency of MoE models by developing a novel parameter-efficient MoE architecture, referred to as SaMoE. SaMoE learns an expert pool that consists of a global set of shared MoE layers and expresses each MoE layer as a soft combination of global MoE layers. Such a scheme decouples the number of experts from MoE model depth, drastically reducing MoE parameters while achieving better accuracy than baseline approaches (Section 4).

**Analysis**. We identify poor gradient flow in MoE layers as the main cause of the poor parameter efficiency of MoE models. Our preliminary analysis shows that expert-sharing helps overcome the poor gradient flow issue and encourages MoE layers to learn more sufficiently (Section 3).

**Evaluation**. We conduct experiments on billion-scale autoregressive MoE language models with open-sourced massive datasets and demonstrate that (i) SaMoE significantly improves the parameter efficiency of MoE models, reducing the model size by up to $5.2\times$ while achieving superior model quality than prior works such as PR-MoE in zero-shot generalization accuracy (Section 5); (ii) Ablation study of the effectiveness of the proposed design elements in SaMoE (Section 5.4); (iii) A detailed evaluation of the scaling properties of SaMoE that reveals the strong scalability of SaMoE (Section 5.5); and (iv) Comparison results between SaMoE and alternative heuristic strategies (Section 5.6). We will also open-source the training and evaluation code at `anonymous_link`.

## 2 RELATED WORK

Mixture-of-Experts architecture converts multiple layers of a deep neural network to sparsely activated counterparts and jointly trained with the rest of the network (Jacobs et al., 1991; Shazeer et al., 2017). It falls into the paradigm of *conditional computation* (Bengio et al., 2013), which was proposed to activate only a small fraction of the model's parameters and computation on-demand on a per-example basis (Bengio et al., 2013; Davis & Arel, 2014; Cho & Bengio, 2014; Bengio et al., 2015). As such, they provide a promising path to build neural networks of much higher capacity without significantly increasing the computation required. Recent work has shown that MoE models can be extended with Transformer architecture for scaling language models(Lepikhin et al., 2020; Fedus et al., 2021). Despite the promising aspects of sparsely activating parameters, MoE models are difficult to train. In particular, prior works attribute the training difficulty of MoE to the unbalanced load of experts and conjecture that encouraging or enforcing all experts to process balanced compute loads can help improve the learning of the gating function. For example, Lepikhin et al. (2020) and Fedus et al. (2021) propose to add load-balancing loss term into the training objective. Lewis et al. (2021) guarantee load balancing across experts by post-processing the routing output to re-assign expert selections to ensure that all experts are selected evenly. Roller et al. (2021) propose to use a fixed hash as the gating function, and Nie et al. (2021) propose to adaptively choose K in top-K selection during training. In addition, another challenge is that the gating function is highly non-differentiable. To address it, Clark et al. (2022) propose to use reinforcement learning for routing, and Hazimeh et al. (2021) learns expert selection through a differentiable loss. Different from those works, our analysis shows that there is parameter redundancy in MoE layers, and we focus on developing parameter-efficient MoE architectures that reduce parameter redundancy.

Parameter efficient architectures have always been an interesting question in machine learning community (Mnih & Hinton, 2008; Mikolov et al., 2010; Press & Wolf, 2017; Inan et al., 2017; Savarese & Maire, 2019; Lan et al., 2020). Lan et al. (2020) discovered that sharing weights across layers improves the parameter efficiency of transformer models (Lan et al., 2020). However, ALBERT focus on encoder-based masked language model pre-training with dense Transformer blocks. More recently, Xue et al. (2022) proposed to share the weights of all MoE layers. However, we find that directly sharing all MoE layers leads to severe accuracy degradation for decoder-based MoE models, especially on large-scale autoregressive GPT-3 style pretraining tasks (e.g., $10$–$100\times$ larger in

model and data size), which motivates the design of SaMoE. Please see our analysis in Section 3 and extensive evaluation in Section 5.

## 3 ANALYSIS OF AUTOREGRESSIVE MoE LANGUAGE MODELS

This section first introduces some preliminaries about MoE and notations needed to describe our approach and then analyzes the behavior of MoE models for language model pretraining.

**Preliminaries.** A Transformer-based MoE model is built on top of a list of $M$ MoE layers extended from a base dense model with $N$-layer ($M \leq N$). Each MoE layer contains a self-attention sublayer and a sparsely activated feed-forward connection (FFC) sub-layer $f_\theta^{(m)}$ that consists $E$ versions of FFCs weights in parallel, where $\theta^{(m)} = \{\theta_1^{(m)}, \theta_2^{(m)}, ..., \theta_E^{(m)}\}$ and $\theta_e^{(m)}$ (i.e., expert) represents the $e$-th version of the parameter ($f_e^m \doteq f_{\theta_e^{(m)}}$) at the $m$-th MoE layer. The number of input tokens to an MoE layer is partitioned $E$-way across experts, and a gating function $g(\cdot) \colon \mathbb{R}^H \to [1, E]^K$ associated with the layer (e.g., a small network) decides which expert(s) an input should be routed to. For example, given an input vector $x_{in} \in \mathbb{R}^H$, where $H$ is the hidden dimension, the gate value of routing $x_{in}$ to experts is: $p_e(x_{in}) = [softmax(g(x_{in}))]_e$. Given the gate values $\{p_e(x_{in})\}_{e=1}^E$, we select the top-K experts to form an activated set of experts $E' \subset \{1...E\}$, where $|E'|$=K. The output $x_{out}$ of the MoE layer is given as the weighted sum of the outputs of each expert: $x_{out} = \sum_{e \in E'} p_e(x_{in}) E_e(x_{in})$. The gate value $p_e(x_{in})$ permits differentiability to the gating function.

In practice, we default $K$=1 because recent studies demonstrated that it leads to a simplified routing strategy while still preserving model quality and reducing communication overhead (Fedus et al., 2021; Clark et al., 2022). Since often $E >> K$ (e.g., K=64), the number of FLOPS required for one input is significantly smaller than the number of parameters in MoE layers. We follow Fedus et al. (2021); Rajbhandari et al. (2022) by placing MoE layers at every other layer. Finally, existing studies identified that balancing workloads across experts is crucial, so we follow prior works to add a load balancing regularization term to make inputs to be more uniformly partitioned across experts (Shazeer et al., 2017; Lepikhin et al., 2020; Fedus et al., 2021). In our experiment, we find that this strategy is sufficient to let each expert take about the same time to finish their assigned load. We note that other load balancing mechanisms could also be adopted (Roller et al., 2021; Lewis et al., 2021; Zuo et al., 2022), which we leave as future work.

**Analysis.** We present several preliminary studies that reveal the challenges of MoE models, which motivate and guide the design in Section 4. We focus on the autoregressive decoder-based MoE: an architecture chosen due to its state-of-the-art performance (Brown et al., 2020; Du et al., 2022; Rajbhandari et al., 2022). Some of the largest models are also decoder-based, making their parameter efficiency important. More details about the experimental setups are presented in Section 5.

**Challenge I: MoEs require significantly more parameters than the base model being extended but exhibit diminishing returns in model quality as we increase the number of experts.** We first investigate the scaling property of autoregressive MoE language models, referred to as AR-MoE, by varying the number of experts per MoE layer from 1 to 64. In all cases, the number of activated parameters per input barely increases due to the sparsity of MoE. Figure 1(a) shows that for training the same number of data samples, adding more experts generally leads to lower validation loss. Meanwhile, while scaling the number of experts brings diminishing returns, the total parameters increase almost exponentially. With E=64, the model size increases from 0.125B to 1.9B, a 15× increase. This observation is consistent with prior work (Fedus et al., 2021; Clark et al., 2022).

**Challenge II: Expert layers have poor gradient flow, receiving insufficient training and failing to contribute to the generalization.** We also study MoE training using gradient flow, which is the first-order approximation of the decrease in the loss expected after a gradient step, to understand how a signal propagates within MoE models. Figure 1(b) shows that the gradients with respect to the weights (measured at 10k steps) are noticeably smaller for MoE layers (layers with even ids). In contrast, we do not observe large unbalanced gradients for non-MoE layers (odd layer ids). During MoE training, the number of input tokens to an MoE layer is partitioned across all experts, so the number of tokens per expert is reduced proportionally to the number of experts compared to the layers where no such partition is done. As such, each expert may receive insufficient training and

become under-fitted. As prior studies suggested that an efficient sparse training regime requires preserving the gradient flow through the network (Wang et al., 2020), we hypothesize that by letting MoE layers receive more gradient flow, the overall parameter efficiency of MoE models can be improved. This observation also indicates that dense layers tend to have a stronger capability to preserve the gradient flow, which may help explain why prior works only extend MoE layers at every other layer (Fedus et al., 2021; Artetxe et al., 2021; Rajbhandari et al., 2022).

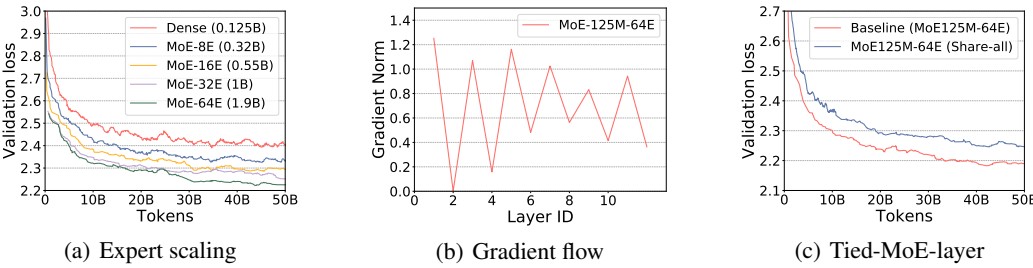

(a) Expert scaling

(b) Gradient flow

(c) Tied-MoE-layer

Figure 1: Autoregressive MoE language model analysis results.

**Challenge III: Sharing the parameter of MoE layers leads to a significant accuracy drop.** Tying weights improves the gradient flow of a layer, because if a weight is used multiple times during training, that weight will receive gradients from multiple places, and those gradients will add up. We experiment with weight sharing on autoregressive MoE models. Unlike the observation in prior works (Dehghani et al., 2019; Lan et al., 2020; Xue et al., 2022), where there is often no accuracy drop when all transformer layers share one set of weights (*share-all*), we find that by tying one MoE layer's weights across all MoE layers, the validation loss suffers significantly, as shown in Figure 1(c). In the evaluation section, we show that the share-all strategy leads to a serious downstream accuracy drop as well, indicating that expert-sharing may not be sufficient for large-scale autoregressive language models (e.g., pre-training GPT-3 style models (Brown et al., 2020)).

# 4    BUILDING PARAMETER-EFFICIENT MOES VIA SAMOE

This section introduces the techniques used in SaMoE to improve the MoE parameter efficiency.

**Self-Adaptive Expert Combinations.** In Transformer-based MoE (Lepikhin et al., 2020) and its variations such as SwitchTransformer (Fedus et al., 2021) and DeepSpeed-MoE (Rajbhandari et al., 2022), each expert layer contains a set of $E$ experts, with no explicit relation between expert parameters of different layers. Conversely, the tied-MoE-layer strategy described in Section 3 has a single set of MoE layers shared among all MoE layers. This strategy leads to accuracy degradation, presumably because the reduced expert capacity hurts the model quality.

While using the same parameters for all MoE layers limits the capability of MoE models, one may wonder if we could relax the technique by dividing MoE layers into groups. As such, instead of preparing parameters for only one MoE layer, we can prepare parameters for $C$ MoE layers to construct a model with $M$ MoE layers, where $1 \leq C \leq M$. For the group's assignment to each layer, we experiment with several strategies and compare them empirically in Section 5.6. Overall, we find that manual determination of an effective group assignment strategy is non-trivial and requires expert knowledge of MoE layer characteristics. It is also not clear that using expert knowledge, even when it is applicable, will lead to superior efficiency. In contrast, we are interested in using a pure learning approach applicable in situations where expert knowledge is unavailable.

Rather than manually determining an expert sharing scheme – we propose to learn an *expert pool* (**EP**), which consists of a *fixed* number of *global* MoE layers $EP=[EP^{(1)}, ..., EP^{(C)}]$, where $C$ is the size of the expert pool (chosen freely as a hyperparameter). Figure 2 illustrates the overview of SaMoE. We then express each MoE layer $f_\theta^{(m)}$ as a linear combination of global MoE layers in the expert pool: $f_\theta^{(m)} := \sum_{c=1}^{C} \alpha_c^{(m)} EP^{(c)}$, where $\alpha_c^{(m)}$, a $C$ dimensional vector, is the coefficient of the $m$-th MoE layer. This scheme allows for coefficient and global MoE layers to be jointly optimized with gradient-based methods. The global MoE layers in the expert pool can be seen as global representation extractors, and coefficient $\alpha^{(m)}$ decides which representations learned by a global layer are relevant for the $m$-th MoE layer computation of a network.

We partition the expert pool across multiple workers, where we place different $EP_m$ to different workers and execute them in parallel in one training step. For an input sequence of $T$ features, we first use $g(\cdot)$ to compute the score for routing each token representation to an $EP_m$. Each worker then sends $T/M$ tokens

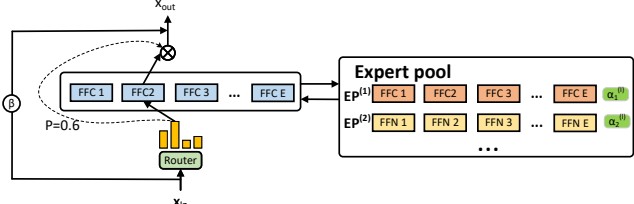

Figure 2: Overview of SaMoE.

to each other worker, with an *all2all* communication operation, similar to standard MoE training. Following prior work (Lan et al., 2020), we make LayerNorm non-shared across MoE layers because (1) they are not parameter and computation intensive, and (2) private LayerNorm encourages more diversified input representations of different MoE layers.

We note that having $C$ as a free parameter decouples the MoE layers in MoE models from the MoE layer depth. Typically, $M$ MoE layers with $E$ experts in each layer and hidden dimension size $H$ have $O(M \cdot E \cdot H^2)$ parameters. With the expert pool, the number of parameters for all MoE layers becomes $O(C \cdot M + C \cdot E \cdot H^2) = O(C \cdot E \cdot H^2)$, where $C \cdot M$ refers to the parameters for coefficient $\alpha$. In practice, we find that increasing the expert pool size leads to better model quality but often a small expert pool size $C = [2, 3]$ already provides similar or better model quality than the baselines.

We also note that in the special case, where the expert pool's cardinality $|C| = 1$, then the proposed scheme turns out to be the tied-MoE-layer strategy described in Section 3. Also, if we constraint coefficient $\alpha^{(m)}$ to be one-hot vectors, SaMoE would make hard sharing assignment similar to manually determined sharing strategy, at the cost of non-differentiability.

Finally, we remark that the idea of learning a combination of experts is inspired by prior studies on *dictionary learning* of sparse coding (Kreutz-Delgado et al., 2003; Mairal et al., 2008; 2009), where a data representation is learned via a linear combination (represented as a sparse vector) of a small set of basic elements (called dictionary). Instead of finding a sparse approximation, we learn expert combinations that improve the expressiveness of global MoE layers.

**Self-Adaptive Shortcut Connections.** While identity mapping has been widely used in dense Transformers for propagating the loss signal across different Transformer layers, Figure 1(b) shows that experts in MoE layers exhibit vanishing gradient phenomenon with identity mapping. Therefore, we conjecture that as we switch to the expert pool, the shortcut connections at those global MoE layers need to be more adaptive as well such that they can expand to cover a sufficiently large number of MoE layers. As such, we apply two additional changes to SaMoE: (1) adding a scaling factor to re-balance the MoE shortcut connection: $x_{out} = \beta x_{in} + f_\theta^{(m)}(LayerNorm(x_{in}))$, where $\beta$ is a learnable parameter and acts as the scaling factor of the shortcut branch, and (2) make each MoE layer has its individual shortcut connection. We empirically find that the self-adaptive shortcut connections help further improve the performance of SaMoE (Section 5.4).

## 5 EXPERIMENTS AND RESULTS

In this section, we first compare SaMoE training to several MoE baselines, to carefully measure the parameter efficiency gains. We then conduct an ablation analysis of SaMoE and evaluate the scaling properties of SaMoE. Finally, we compare SaMoE with several heuristic-based sharing strategies.

### 5.1 EXPERIMENTAL SETUP

**Data.** We follow recent practice (Black et al., 2022; Rae et al., 2021) and use the public available Pile dataset (Gao et al., 2020) to train the autoregressive language models. Pile is a massive curated dataset specifically designed for training large language models. It contains 825GiB English text corpus composed of 22 diverse web-domain datasets and has been demonstrated to achieve comparable performance to similarly sized GPT-3 models. Following Brown et al. (2020), we train all configurations for 300 billion tokens for fair comparison unless otherwise indicated. For downstream task evaluation, we follow prior work Rajbhandari et al. (2022) and primarily focus on zero-shot generalization. As described in Brown et al. (2020), zero-shot evaluation is one of the most challenging settings, because the model is only given a natural language instruction describing the task and has

been used by prior work for monitoring scientific progress (Rajbhandari et al., 2022). We use the Eleuther AI Language Model Evaluation Harness (Gao et al., 2021), an open-sourced codebase for downstream task zero-shot evaluation, covering a wide range of NLP tasks.

**Models.** We focus our evaluation on autoregressive MoE models whose base architecture mainly follows that of GPT-3 (Brown et al., 2020). As the main goal of the evaluation is to show that our proposed method can improve the parameter efficiency of MoE models instead of achieving SoTA performance, we choose a model that can finish the training in a reasonable time frame (e.g. in roughly 120 wall-clock hours on 32 A100 GPUs for one run). As such, we choose a 1.9 billion-parameter MoE model using a 125M dense model (12 layers, 768 hidden sizes, 12 attention heads) the same as the GPT-3 small setting in Brown et al. (2020), as the base model. Then we add E=64 MoE layers on every other FFC layer. We choose E=64 by following Clark et al. (2022); Rajbhandari et al. (2022), which suggest targeting E∈{64, 128}.

**Hyperparameters.** We largely follow the methodology from Brown et al. (2020) and Rajbhandari et al. (2022) to train models. For detailed hyperparameter settings, please refer to Appendix A.

**Framework.** We implement and train our models in PyTorch (Paszke et al., 2019) using DeepSpeed library (Rajbhandari et al., 2022). We provide more implementation details in Appendix B.

**Compared Models.** We compare the proposed strategy with the following schemes:

- **AR-MoE**: The first baseline is an autoregressive MoE model, which is a decoder-based MoE model, as described in Rajbhandari et al. (2022).
- **PR-MoE**: We follow the training procedure in Rajbhandari et al. (2022) to train a PR-MoE by using half experts for all MoE layers except the last two MoE layers.
- **Grouped-MoE**: We create a strong baseline by partitioning MoE layers into $C$ groups (e.g., *1g*, *2g*, *3g*), and let every $\lfloor M/C \rfloor$ consecutive MoE layers share their parameters.
- **SaMoE**: The proposed method described in Section 4.

## 5.2 Pre-training with SaMoE

To monitor the impact of SaMoE on the pre-training, we report the validation loss during pre-training because it shows how SaMoE affects the convergence of the pre-training of the autoregressive MoE model in a straightforward way. Figure 3 shows that despite having a smaller model size, SaMoE outperforms AR-MoE, PR-MoE, and Grouped-MoE by a wide margin. PR-MoE achieves a similar validation loss as AR-MoE but is worse than SaMoE. The primary difference between PR-MoE and SaMoE is that PR-MoE exploits layer sensitivity and uses half experts for lower MoE layers, which reduces the number of experts by less than $2\times$. In contrast, SaMoE decouples the number of experts from the model depth and allows MoE models to scale to deeper models without significantly increasing the number of parameters. SaMoE further improves the parameter efficiency of the remaining parameters via self-adaptive expert combinations. A clear goal for future work includes combining SaMoE and heterogeneous experts to achieve even greater

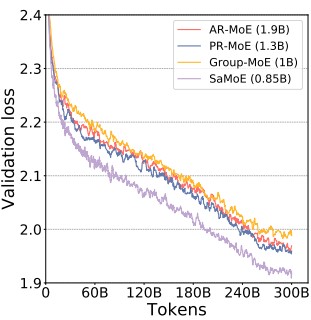

Figure 3: The validation loss comparison results of MoE, PR-MoE, Group-MoE, and SaMoE.

parameter efficiency of MoE models. Grouped-MoE (3 groups) struggles to achieve comparable validation loss as AR-MoE despite already having a $1.4\times$ larger model size than SaMoE, indicating that sharing weights via grouped MoE layers is insufficient to achieve high MoE model quality. Further increasing the group size helps mitigate the accuracy gap, at the cost of significantly more trainable parameters (as analyzed in Section 5.6). Finally, since AR-MoE, PR-MoE, and SaMoE all have converged, it would be hard for them to outperform SaMoE on validation loss even if we spent longer training time. Therefore, this figure indicates that our proposed method is more parameter-efficient than baselines for pre-training tasks.

## 5.3 Downstream Evaluation

Table 1 showcases the most important advantage of SaMoE's design choices via zero-shot evaluation. We provide full results of 21 tasks in Appendix C. The first row is an MoE model with a

comparable model size from related work trained with a different dataset (Du et al., 2022). We make several observations: (1) with only around 37% of AR-MoE's parameters (0.7B vs. 1.9B), SaMoE achieves comparable accuracy as the baseline AR-MoE, e.g., 40 vs. 39.9 on average across 21 tasks. (2) By increasing the depth of the MoE layer, we obtain a SaMoE model that has 0.85B parameters (44% of AR-MoE) but achieves significantly zero-shot evaluation improvements over AR-MoE, as measured by the difference of a wide range of downstream tasks: Lambada (+3 points), Piqa (+1.5 points), Copa (+2 points), RTE (+2.5 points), Wikitext (+1.3 points) and on average 0.9 points across 21 tasks. SaMoE achieves much better parameter efficiency than AR-MoE in that instead of relying on parameter-intensive MoE layers at every other Transformer layer, SaMoE learns an expert pool of global MoE layers and letting each MoE layer learns a linear combination of the global MoE layers in the pool. More importantly, the number of parameters does not increase as we scale the model depth, improving the parameter efficiency while drastically reducing the model size.

Table 1: Zero-shot evaluation comparison between AR-MoE, PR-MoE, Grouped-MoE, and SaMoE.

| Model #params | | Lambada | Piqa | Copa | RTE | Avg. (21 tasks) (↑) | WikiText (↓) | Speedup |
|---|---|---|---|---|---|---|---|---|
| 100M-64E (Du et al., 2022) | 1.9B | 36.9 | 69.0 | 53.6 | 29.1 | N/A | N/A | N/A |
| AR-MoE (125M-64E) | 1.9B | 48 | 67.7 | 67 | 54.2 | 40.2 | 20.1 | 1× |
| PR-MoE (125M-32/64E) | 1.3B | 33.8 | 63.2 | 68 | 52 | 38.3 | 34.0 | 1.05× |
| Grouped-MoE: | | | | | | | | |
| 125M-64E-1g | 0.42B | 44.1 | 64 | 65 | 53.1 | 38.9 | 23.9 | 1× |
| 125M-64E-2g | 0.7B | 46.2 | 66.9 | 67.0 | 54.9 | 39.6 | 21.9 | 1× |
| 125M-64E-3g | 1B | 48.2 | 67.7 | 67.0 | 57.8 | 39.7 | 21.1 | 1× |
| SaMoE: | | | | | | | | |
| 125M-64E-12L | **0.72B** | 48.1 | 66.5 | 67 | 52.7 | 40.2 | 21.2 | 0.73× |
| 125M-64E-24L | **0.85B** | 51 | 68.5 | 69 | 56.7 | **41.1** | **18.8** | 0.36× |

Compared with PR-MoE, SaMoE achieves 2.3 points higher zero-shot generalization accuracy on average with only 69% of its parameters. We note that PR-MoE incurs a slightly larger accuracy loss over AR-MoE because the original PR-MoE used a different learning rate from AR-MoE. In contrast, in our experiments, all configurations use the same learning rate for a fair comparison. Furthermore, PR-MoE achieves more comparable accuracy to AR-MoE when the base model becomes larger, as shown in the next part. It is possible that by slightly tuning the hyperparameters of PR-MoE, its zero-shot evaluation results can be improved. SaMoE achieves higher parameter efficiency than PR-MoE because SaMoE decouples the expert parameters from the MoE model depth, leading to a higher parameter reduction ratio than PR-MoE. SaMoE achieves +1.4 points higher average accuracy than Grouped-MoE (41.1 vs. 39.7) and +2.3 points better perplexity on WikiText despite having 89% of parameters (0.89B vs. 1B). As expected, the number of parameters of Grouped-MoE decreases as we decrease the number of groups in Grouped-MoE. However, we observe that there are also drastic zero-shot evaluation accuracy decreases as we do so. As an example, the LAMBADA accuracy drops 4.1 points when switching from Grouped-MoE (3g) to Grouped-MoE (1g). Therefore, Grouped-MoE is not sufficient to achieve parameter efficient MoE for autoregressive language models. These results demonstrate the parameter efficiency benefits of SaMoE.

## 5.4 ABLATION AND ANALYSIS RESULTS

In this part, we study the importance of each component of SaMoE: self-adaptive expert combination (SEC), and self-adaptive shortcut connection (SSC). We use 125M-64E MoE models in this study. As such, *SaMoE- SSC* represents disabling self-adaptive shortcut connection. *SaMoE- SEC - SSC* further disables adaptive expert combination so the model uses hard sharing strategies for MoE layers. Furthermore, we also include SaMoE with only one global MoE layer, which removes group sharing (GS) completely (*SaMoE- GS - SEC - SSC*) as an additional baseline. Figure 6 shows the validation perplexity of different configurations. It is expected that the removal of either component in SaMoE results in a performance drop.

## 5.5 EVALUATION OF THE SCALING PROPERTY OF SAMOE

In this part, we investigate the scaling properties of SaMoE. As scaling billion-scale MoE models often require significant training budgets, we choose the number of updates so that each training can still be complete in roughly 72 wall-clock hours.

**Scaling in Expert Pool Size and Width.** Expressing an MoE layer as a linear combination of global MoE layers improves the parameter efficiency. Of immediate interest is $C$: the expert pool size. A larger $C$ increases the number of parameters in the model but also enables a more flexible combination of global experts. To verify if a larger expert pool size helps improve the MoE model quality, we compare SaMoE with $C = 2$ and $C = 3$. We also study SaMoE with different expert widths (e.g., H=768, H=1024). Figure 4(a) shows the results. We make several observations: (1) In all cases, SaMoE consistently outperforms the baseline despite a smaller model size, indicating that scaling expert pool size and width brings additional benefits in model convergence while still allowing SaMoE to be more parameter efficient than the baseline. (2) Increasing $C$ increases the model size by 14–17%, depending on the hidden dimension. However, by doing so, the validation perplexity also gains noticeable improvements (e.g., ∼0.25 points). (3) Scaling either $C$ or $H$ improves the convergence, though scaling $H$ together with a smaller $C$=[2,3] appears more competitive.

**Scaling in Model Depth and Experts.** Given that SaMoE achieves higher accuracy than alternative methods with fewer parameters, one interesting question is *how would SaMoE scale as we further increase the depth of the model and increase the number of experts?* To study this, we investigate three configurations: (1) *scaling M*: varying the depth of the model from 12L to 24L (L24, E=64), (2) *scaling E*: varying the number of experts from 64 to 128 (L12, E=128), and (3) *scaling both M and E*: increasing both the depth and the number of experts (L24, E=128). In all three cases, we keep the expert pool size $C$ to the same. Figure 4(b) shows the scaling results. We observe that *increasing both model depth and experts leads to further improvement of SaMoE model equality, and compound benefits can be achieved by scaling along both dimensions.* However, while scaling $E$ leads to an increased number of parameters and model sizes, scaling along the depth dimension $M$ leads to improvements with only a mild increase (+18%) in model size. Therefore, scaling the depth dimension of SaMoE appears to be more effective from a parameter efficiency perspective.

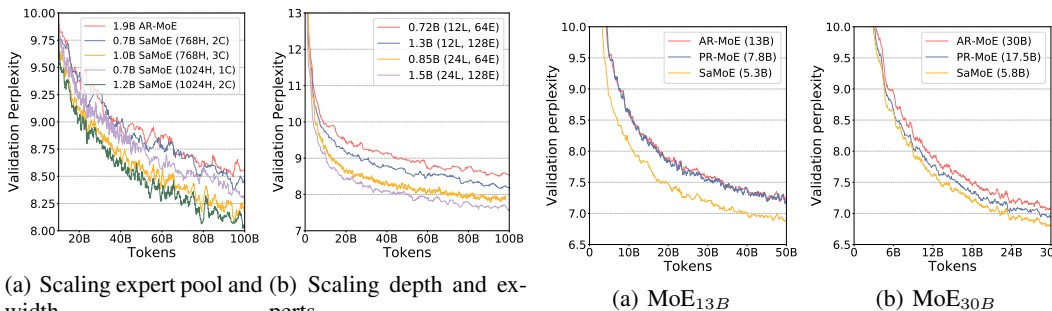

(a) Scaling expert pool and width  (b) Scaling depth and experts

Figure 4: Evaluation results of scaling SaMoE by varying the expert pool size, depth, and experts.

(a) MoE$_{13B}$  (b) MoE$_{30B}$

Figure 5: Comparison results to MoE$_{13B}$ and MoE$_{30B}$.

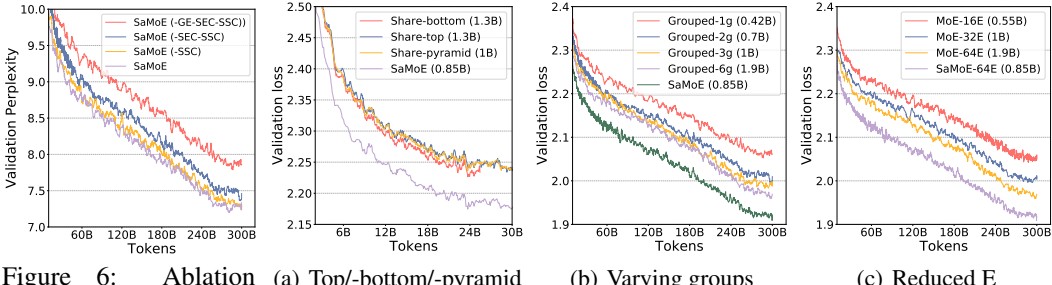

Figure 6: Ablation study results of SaMoE.

(a) Top/-bottom/-pyramid  (b) Varying groups  (c) Reduced E

Figure 7: Analysis and comparison results of different MoE architectures.

**Larger MoE models.** We evaluate SaMoE on two larger MoE models, including (1) 13B/350M+128E: a 13 billion-parameter MoE model that uses 350M as the base model and with 12 128-expert MoE layers, and (2) 30B/760M+128E: a 30 billion-parameter MoE model with 760M as the base model and 12 128-expert MoE layers. For (1), we scale SaMoE along the width dimension (1024→1536), which produces a 5.3B model; and for (2), we scale SaMoE along the depth dimension (24→32), which leads to a 5.8B parameter model. Due to resource limitations, we train 100k and 50k steps (out of 572k total steps) for the 13B and 30B models, respectively. Figure 5(a)

and Figure 5(b) show the validation loss of these two models. SaMoE outperforms the AR-MoE baseline by a large margin with $2.5\times$ fewer parameters in comparison to the 13B baseline model. For the 30B model, SaMoE achieves slightly better validation loss as the baseline with $5.2\times$ model size reduction. Table 2 also confirms that the same trend is observed on zero-shot evaluation. In Appendix C, we also provide the complete zero-shot evaluation of these two models.

Table 2: Zero-shot evaluation results of 350M-128E and 760-128E MoE models.

| Tasks | 350M-128E | | | 760M-128E | | |
|---|---|---|---|---|---|---|
| | AR-MoE | PR-MoE | SaMoE | AR-MoE | PR-MoE | SaMoE |
| #Params | 13B | 7.6B | **5.3B** | 30B | 17.5B | **5.8B** |
| lambada | 47.2 | 49.4 | **52.4** | 44.8 | 47.2 | **48.3** |
| piqa | 68.2 | 66.5 | **68.9** | 67.8 | 67.0 | 67.6 |
| race | 29.8 | 30.6 | **32.6** | 30.4 | **31.0** | 29.9 |
| copa | 66.0 | **69.0** | 68.0 | 62.0 | 68.0 | **70.0** |
| Avg Acc. (20 tasks) | 40.7 | 41.1 | **41.6** | 41.2 | 41.4 | **41.7** |

## 5.6 EVALUATION OF ALTERNATIVE STRATEGIES

As described in Section 3, there are other strategies for sharing MoE layer parameters. As such, we have also explored and quantified alternative cross-sharing strategies for building MoE.

**Share-bottom/-top vs. SaMoE**. *Share-bottom* shares the parameter of MoE layer with the first half MoE layers and leaves the second half of MoE layers unshared, and *Share-top* switches the shared MoE layer to the second half of the layers. Although both strategies have the same model size, Figure 7(a) shows that *share-bottom* leads to lower validation loss than *Share-top*. This is somewhat expected. From Figure 1(b), we see that higher MoE layers have larger gradient norms, which implies that higher MoE layers tend to have more freedom than lower layers for their expressiveness. In other words, lower MoE layers probably have more redundant parameters compared to the higher MoE layers. That said, *share-bottom* still has the other half of MoE layers non-shared. Therefore, its model size is still quite large compared to SaMoE (1.3B vs.0.89B).

**Exploiting MoE layer heterogeneity**. Rajbhandari et al. (2022) reported heterogeneity in MoE layers, e.g., the last two MoE layers are more important than the remaining ones and should contain more experts. To investigate how layer heterogeneity affects expert sharing, we investigate another strategy, referred to as *Share-pyramid*, which lets the last several MoE layers (e.g., 2) have their own weights while the other MoE layers share parameters. Figure 7(a) also shows that while *Share-pyramid* achieves similar convergence as *Share-top* with a smaller model size. However, compared with SaMoE, layer heterogeneity has a very minimal effect on model convergence.

**MoE Layer sharing with varying groups**. Figure 7(b) shows that by increasing the groups in *Grouped-MoE* ($1g \rightarrow 2g \rightarrow 3g$), the validation loss decreases. However, even with 3 groups, the Grouped-MoE cannot obtain the same validation loss as the baseline model and SaMoE despite having a larger model size.

**What if we train an MoE with a smaller number of experts?** Another method to obtain an MoE model with a smaller number of parameters is to reduce the number of experts. Figure 7(c) shows that although reducing the number of experts per MoE layer (e.g., 64→32→16) reduces the model size, there is a significant drop in model quality. As such, our proposed method is much more parameter efficient than models with reduced $E$.

## 6 CONCLUSION AND FUTURE DIRECTIONS

This work investigates parameter-efficient architecture design for Mixture-of-Expert models. Based on the analysis of the challenges of MoE, we introduce a novel MoE method called SaMoE that learns self-adaptive expert combinations for each MoE layer. Our extensive experiments demonstrate that our SaMoE is more parameter-efficient than prior arts and can significantly reduce the model size while achieving high accuracy. While this work mainly focuses on the autoregressive MoE models, future work might investigate the performance of our methods on other tasks and MoE architectures. Another exciting topic is studying the best trade-off between scaling in expert pool size, depth, and experts.

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

In this part, we present results that are not included in the main text due to the space limit.

## A  HYPERPARAMETERS

In this part, we include detailed hyperparameters used for training the models in this work. We largely follow prior works (Brown et al., 2020; Rajbhandari et al., 2022) to set the hyperparameters. The batch size and learning rates are set according to the model size following Brown et al. (2020) (e.g., batch size 256, learning rates 6e-4). We linearly warm-up the learning rate from a min learning rate (6e-5) over the first 375M tokens and use a cosine learning rate decay back to min lr over the remaining tokens. For all experiments, we use a batch size of 4 for each GPU, with gradient accumulation of 2, and train with FP16. Table 3 provides the detailed hyperparameters used for training models in Section 5.2 and Section 5.3. Table 4 provides the hyperparameters used for training 13B MoE and 30B MoE, respectively, in Section 5.5.

Table 3: Sizes, architectures, and learning hyperparameters of the MoE125M-64E MoE models we trained.

|  | AR-MoE | PR-MoE | Grouped-MoE | S-MoE-1 | S-MoE-2 |
|---|---|---|---|---|---|
| Num. parameters | 1.9B | 1.3B | 1B | 0.72B | 0.85B |
| Num. layers | 12 | 12 | 12 | 12 | 24 |
| Hidden size | 768 | 768 | 768 | 768 | 768 |
| Num. attention heads | 12 | 12 | 12 | 12 | 12 |
| Num. experts per layer | 64 | 32/64 | 64 | 64 | 64 |
| Num. global MoE layers | N/A | N/A | 3 | 2 | 2 |
| Context/sequence length | 2K | 2K | 2K | 2K | 2K |
| Training tokens | 300B | 300B | 300B | 300B | 300B |
| Batch size | 256 | 256 | 256 | 256 | 256 |
| Learning rate | 4.50E-04 | 4.50E-04 | 4.50E-04 | 4.50E-04 | 4.50E-04 |
| Min. learning rate | 4.50E-06 | 4.50E-06 | 4.50E-06 | 4.50E-06 | 4.50E-06 |
| LR linear warmup tokens | 375M | 375M | 375M | 375M | 375M |
| LR cosine decay tokens | 300B | 300B | 300B | 300B | 300B |
| Init standard deviation | 0.014 | 0.014 | 0.014 | 0.014 | 0.014 |
| Adam beta1 | 0.9 | 0.9 | 0.9 | 0.9 | 0.9 |
| Adam beta2 | 0.95 | 0.95 | 0.95 | 0.95 | 0.95 |
| Weight decay | 0.1 | 0.1 | 0.1 | 0.1 | 0.1 |
| Grad clip | 1 | 1 | 1 | 1 | 1 |
| MoE loss coefficient | 0.01 | 0.01 | 0.01 | 0.01 | 0.01 |

Table 4: Sizes, architectures, and learning hyperparameters of the MoE350M-128E and MoE760M-128E models.

|  | 350M-128E | | | 760M-128E | | |
|---|---|---|---|---|---|---|
|  | AR-MoE | PR-MoE | SaMoE | AR-MoE | PR-MoE | SaMoE |
| Num. parameters | 13B | 7.6B | 5.3B | 30B | 17.5B | 5.8B |
| Num. layers | 24 | 24 | 24 | 24 | 24 | 32 |
| Hidden size | 1024 | 1024 | 1536 | 1536 | 1536 | 1536 |
| Num. attention heads | 16 | 16 | 16 | 16 | 16 | 16 |
| Num. experts per layer | 128 | 64/128 | 128 | 128 | 64/128 | 128 |
| Num. global MoE layers | N/A | N/A | 2 | N/A | N/A | 2 |
| Context/sequence length | 2K | 2K | 2K | 2K | 2K | 2K |
| Training tokens | 50B | 50B | 50B | 25B | 25B | 25B |
| Batch size | 256 | 256 | 256 | 256 | 256 | 256 |
| Learning rate | 2.00E-04 | 2.00E-04 | 2.00E-04 | 1.80E-04 | 1.80E-04 | 1.80E-04 |
| Min. learning rate | 2.00E-06 | 2.00E-06 | 2.00E-06 | 1.00E-06 | 1.00E-06 | 1.00E-06 |
| LR linear warmup tokens | 375M | 375M | 375M | 375M | 375M | 375M |
| LR cosine decay tokens | 300B | 300B | 300B | 300B | 300B | 300B |
| Init standard deviation | 0.014 | 0.014 | 0.014 | 0.014 | 0.014 | 0.014 |
| Adam beta1 | 0.9 | 0.9 | 0.9 | 0.9 | 0.9 | 0.9 |
| Adam beta2 | 0.95 | 0.95 | 0.95 | 0.95 | 0.95 | 0.95 |
| Weight decay | 0.1 | 0.1 | 0.1 | 0.1 | 0.1 | 0.1 |
| Grad clip | 1 | 1 | 1 | 1 | 1 | 1 |
| MoE loss coefficient | 0.01 | 0.01 | 0.01 | 0.01 | 0.01 | 0.01 |

## B  IMPLEMENTATION DETAILS

We implement and train our models in PyTorch (Paszke et al., 2019) using DeepSpeed library (Rajbhandari et al., 2022). To implement SaMoE, we partition the global expert layers and their computation using expert parallelism, where we place different experts on different devices and execute them in parallel in one mini-batch training step. We use the All-to-All collective to handle the communication among experts. When the number of experts in the base layer is smaller than the number of available GPUs, we use expert parallelism in combination with data parallelism as described in (Rajbhandari et al., 2022). We use data parallelism for non-MoE layers.

We use a top-K gating function to activate a subset of K experts in the MoE layer for each token. Specially, we follow Fedus et al. (2021) to choose top-1 expert selection. As such, the MoE model has roughly the same number of parameters to be activated for each token as its dense counterpart. During training and inference, each expert has a capacity of $min(min\_capacity, \lceil \frac{Q \cdot B}{E} \rceil)$ tokens, which determine how many tokens a single expert can handle. $Q$ is a capacity factor which we set to 1.0, B is the total number of tokens in a batch, and we set min_capacity to 4. We handle tokens that exceed the capacity of an expert the same way as Fedus et al. (2021), by considering additional tokens as "overflowed" and passing their representations to the next layer via the shortcut connection.

**Hardware.**  We conduct the pre-training and evaluation with 32 NVIDIA Ampere A100-40GB GPUs.

## C  ZERO-SHOT EVALUATION RESULTS

This part includes the complete zero-shot evaluation results on 21 downstream tasks.

Table 5: Zero-shot evaluation results of baselines and SaMoE on 125M-64E MoE models.

| | AR-MoE | PR-MoE | Grouped-MoE | | | SaMoE | |
|---|---|---|---|---|---|---|---|
| | | | 1g | 2g | 3g | L=12 | L=24 |
| Tasks — #Parmas | 1.9B | 1.3B | 0.42B | 0.72B | 1B | 0.72B | 0.85B |
| lambada | 48.0 | 33.8 | 44.1 | 46.2 | 48.2 | 48.1 | 51.3 |
| triviaqa | 3.6 | 1.8 | 2.4 | 2.6 | 3.0 | 2.9 | 3.8 |
| webqs | 2.0 | 1.3 | 1.4 | 1.4 | 1.6 | 1.4 | 0.9 |
| piqa | 67.7 | 63.2 | 64.0 | 66.9 | 67.7 | 66.5 | 68.6 |
| race | 30.3 | 28.2 | 30.5 | 30.1 | 29.4 | 31.9 | 31.4 |
| boolq | 55.0 | 55.4 | 58.2 | 59.2 | 54.2 | 59.4 | 55.7 |
| copa | 67.0 | 68.0 | 65.0 | 67.0 | 67.0 | 67.0 | 69.0 |
| winogrande | 51.9 | 52.1 | 50.3 | 50.0 | 52.2 | 50.9 | 51.6 |
| arc_challenge | 25.0 | 23.5 | 23.6 | 21.6 | 24.5 | 25.1 | 26.3 |
| arc_easy | 45.7 | 41.3 | 42.0 | 44.1 | 44.6 | 45.3 | 47.3 |
| openbookqa | 30.2 | 28.8 | 28.4 | 17.4 | 29.4 | 30.6 | 30.8 |
| cb | 33.9 | 33.9 | 23.2 | 37.5 | 25.0 | 28.6 | 28.6 |
| rte | 54.2 | 52.0 | 53.1 | 54.9 | 57.8 | 52.7 | 56.7 |
| wic | 50.0 | 49.4 | 50.0 | 50.0 | 49.7 | 50.0 | 50.0 |
| wsc | 36.5 | 42.3 | 36.5 | 36.5 | 36.5 | 36.5 | 37.5 |
| multirc | 0.8 | 1.9 | 0.8 | 0.8 | 0.9 | 0.9 | 0.8 |
| record-f1 | 73.7 | 63.1 | 71.9 | 72.8 | 73.8 | 73.8 | 76.5 |
| record-em | 73.1 | 62.4 | 71.2 | 72.0 | 73.1 | 73.1 | 75.8 |
| anli_r1 | 31.9 | 34.2 | 34.9 | 33.1 | 31.2 | 32.9 | 33.5 |
| anli_r2 | 31.6 | 33.1 | 32.9 | 34.4 | 31.0 | 33.2 | 34.0 |
| anli_r3 | 32.1 | 33.8 | 33.3 | 33.6 | 32.8 | 33.7 | 33.0 |
| AVG Acc. | 40.2 | 38.3 | 38.9 | 39.6 | 39.7 | 40.2 | **41.1** |

Table 6: Zero-shot evaluation results of baselines and SaMoE on 350M-128E and 760-128E MoE models.

| Tasks | 350M-128E | | | 760M-128E | | |
|---|---|---|---|---|---|---|
| | AR-MoE | PR-MoE | SaMoE | AR-MoE | PR-MoE | SaMoE |
| #Params | 13B | 6.7B | **4.8B** | 30B | 17.5B | **5.8B** |
| lambada | 47.2 | 49.4 | 52.4 | 44.8 | 47.2 | 48.3 |
| triviaqa | 4.6 | 2.4 | 4.3 | 2.8 | 2.5 | 3.9 |
| webqs | 1.8 | 1.4 | 0.9 | 1.2 | 0.7 | 1.0 |
| piqa | 68.2 | 66.5 | 68.9 | 67.8 | 67.0 | 67.6 |
| race | 29.8 | 30.6 | 32.6 | 30.4 | 31.0 | 29.9 |
| boolq | 59.0 | 55.6 | 56.4 | 56.9 | 61.3 | 60.6 |
| copa | 66.0 | 69.0 | 68.0 | 62.0 | 68.0 | 70.0 |
| winogrande | 50.4 | 51.5 | 53.4 | 53.3 | 52.6 | 51.4 |
| arc_challenge | 25.9 | 23.2 | 24.8 | 24.8 | 25.3 | 26.1 |
| arc_easy | 46.6 | 52.6 | 52.9 | 51.8 | 52.4 | 51.8 |
| openbookqa | 31.6 | 29.6 | 31.0 | 29.0 | 29.0 | 29.0 |
| copa | 66.0 | 71.0 | 68.0 | 58.1 | 54.2 | 50.2 |
| rte | 53.8 | 53.4 | 52.7 | 50.0 | 50.0 | 49.7 |
| wic | 49.7 | 50.0 | 50.3 | 41.3 | 36.5 | 37.5 |
| wsc | 37.5 | 36.5 | 36.5 | 1.8 | 1.8 | 0.8 |
| multirc | 1.0 | 1.3 | 1.7 | 74.1 | 74.7 | 75.3 |
| record | 75.0 | 75.4 | 77.9 | 73.4 | 74.0 | 74.5 |
| anli_r1 | 31.8 | 34.2 | 32.3 | 31.8 | 33.5 | 34.5 |
| anli_r2 | 33.3 | 33.1 | 33.3 | 32.1 | 32.9 | 35.3 |
| anli_r3 | 34.8 | 34.9 | 34.4 | 36.5 | 33.8 | 35.8 |
| AVG Acc. | 40.7 | 41.1 | **41.6** | 41.2 | 41.4 | **41.7** |

