# OpenReview forum: "SaMoE: Parameter Efficient MoE Language Models via Self-Adaptive Expert Combination"
_ICLR.cc/2023/Conference — Submitted to ICLR 2023_

### Official Review · Reviewer_9vyj · 2022-10-25

**Confidence:** 5
**Correctness:** 2
**Technical Novelty And Significance:** 2
**Empirical Novelty And Significance:** 2
**Recommendation:** 3

**Clarity, Quality, Novelty And Reproducibility:**

Quality:

Due to the above weaknesses, this paper does not have a high quality.


**Strength And Weaknesses:**

Weaknesses:

(1). This paper carried out analysis first and listed three challenges from analysis. However, I did not know which MoE model does this paper study. In Figure 1, it shows “MoE” but I don’t know which MoE model is used to carry out experiments. There are plenty of MoE models such as Gshard (Lepikhin et al., 2020), Switch Transformer (Fedus et al., 2021), Base Layers (Lewis et al., 2021), HASH Layers (Roller et al., 2021), and etc. Different MoE models may lead to different conclusions. The author needs to announce which model they used for analysis and add citations.

(2). This article used only one MoE model to draw analysis conclusions, which I cannot agree with. Because different MoE models may have different performance, analysis conclusions need to conduct experiments with at least two representative MoE models when talking about common challenges with MoE models.

(3). I am very suspicious about the expert pool method proposed in this article. How to choose the size of the expert pool. I speculate that the amount of experts required by a MoE model may be related to the diversity of the dataset. Table 1 in BASE layers paper [1] shows similar words usually gathered to the same expert unit. However, this article only uses one dataset for pretraining, and does not use multiple datasets to test the required expert pool size.

(4). Followed by the third problem, this paper selected the Pile dataset as the pre-training dataset. However, the Pile dataset is full of duplicate documents (see [2] page 2), and this paper does not perform additional de-duplication processing. Because the dataset selected in the article has a lot of repetition and the tokens are not diverse, the size of the expert pool does not need to be large. The conclusion is likely to change when changing to a different (diverse) pre-training dataset.

(5). As a MoE model, it is basically necessary to control the number of flos and compare it with the dense models and sparse models with the same number of flops, but this paper does not report total training flops number and total train computer (PF-days). In addition, this paper doesn’t compare it with a dense model with the same amount of flops in table 1.

(6). I also have some questions about the experimental results of table 2. When we compared SaMoE (350M-128E) with dense model (350M), SaMoE should have more flops since it needs additional all2all communication cost. However, I notice usually a dense model (350M) could get a score of 70.2 on piqa. This SaMoE with more flops achieves a score 68.9.

(7). Minor suggestion: usually we reported pretraining perplexity instead of validation loss in figure 3.


*References:*

[1]. Lewis, Mike, et al. "Base layers: Simplifying training of large, sparse models." International Conference on Machine Learning. PMLR, 2021.

[2]. Zhang, Susan, et al. "Opt: Open pre-trained transformer language models." arXiv preprint arXiv:2205.01068 (2022).


**Summary Of The Paper:**

 This paper proposes a new MoE model Architecture to improve the parameter efficiency of MoE by learning a soft combination of a global set of expert layers.


**Summary Of The Review:**

This paper proposes a new MoE model. However, in the analysis part, it only carried out analysis experiments with one MoE model, which is hard to tell if findings applied to all MoE models. In addition, this paper proposes to have a fixed number of global MoE layers, which is probably not suitable when a pre-training dataset has very diverse tokens. It happens that this paper selects the Pile as the pretraining dataset, and Pile is widely considered to contain many repeated sentences. (see [2] page 2).

*References:*

[1]. Lewis, Mike, et al. "Base layers: Simplifying training of large, sparse models." International Conference on Machine Learning. PMLR, 2021.

[2]. Zhang, Susan, et al. "Opt: Open pre-trained transformer language models." arXiv preprint arXiv:2205.01068 (2022).

---

> ### Author Response · Authors · 2022-11-16
> **Response to Reviewer 9vyj (Part 1)**
>
> ***\* This paper carried out analysis first and listed three challenges from analysis. However, I did not know which MoE model does this paper study…. The author needs to announce which model they used for analysis and add citations.\****
>
> **A**: For the analysis, we use autoregressive MoE from DeepSpeed-MoE [1] as a baseline to conduct the analysis in Section 3, which is a decoder-based MoE model. We thought this was clear because right before Section 3 and in the last sentence of Section 2, we explicitly said we studied "autoregressive GPT-3 style MoE" and "see our analysis in Section 3", and again in the fourth paragraph of Section 3, right before we describe the analysis results, we mentioned that "We focus on the autoregressive decoder-based MoE: an architecture chosen due to its state-of-the-art performance {citations}" with proper citations. We now realize that this might still cause some confusion. We will try to highlight it even more in Section 3.
>
> ***\* This article used only one MoE model to draw analysis conclusions, which I cannot agree with.\****
>
> **A**: We mentioned in Section 3 (in the section title as well as in the fourth paragraph) that our results are observed from autoregressive decoder-based MoE models (not any MoE models), which motivate the design in Section 4. We also note that given the high training cost of MoE, prior studies often focus on one type of MoE model. For instance, GLaM [2] focuses on decoder-only MoE, SwitchTransformer[3] and GShard[4] focus on encoder-decoder MoE, and Unified Scaling Law [5] studies only decoder-only MoE. We suspect that the reason is that given the billion-scale parameters and high training cost, having a wide coverage of MoE models is very challenging. That said, we believe that models at different scales may exhibit different phenomena. That's why in Section 5, we include studies of \textbf{three autoregressive MoEs} of different scales ranging from 1.9-30 billion parameters. We will further highlight that our paper focused on autoregressive MoE models.
>
> ***\* I am very suspicious about the expert pool method proposed in this article. How to choose the size of the expert pool. \****
>
> **A**: Our claim about our proposed expert pool method is supported by extensive evidence in Section 5. In Section 5.5, we provided an analysis of "Scaling in Expert Pool Size," which shows results of varying expert pool sizes. Overall, the expert pool size is a hyperparameter that controls the number of parameters and model convergence quality. The smaller the expert pool size, the more parameter-efficient SaMoE is, but it is also more likely that accuracy could suffer. To tune this hyperparameter, we suggest starting from 1 and increasing it linearly (like, 2, 3, 4), until a satisfactory accuracy is reached. In our experiments, we start with 2 (2 x 64 = 128 and 2 x 128 = 256 experts in total for the 1.9B MoE model and 30B MoE, respectively). We choose an expert pool size where the training curve matches or even outperforms the original baseline (often by looking at the training curve from the first few thousand steps).
>
> ***\* However, this article only uses one dataset for pretraining, and does not use multiple datasets to test the required expert pool size. \****
>
> **A**: It is standard in GPT-style pre-training literature to use one giant dataset that is a union of diverse datasets, e.g., [1, 2,3,5,6,7, 8, ]. Given that PILE is already a union of 22 diverse datasets and one of the largest open datasets, we thought it was a challenging dataset to test the proposed expert pool method. We also trained the model for 300 billion tokens, which is similar to GPT-3. Finally, decreasing the size of the dataset (e.g., using a subset of PILE) will decrease the model-vs-data ratio, which results in more expert redundancy, so it is more likely that a small expert pool size is sufficient to obtain on-par model quality, and vice versa.

---

> > ### Author Response · Authors · 2022-11-16
> > **Response to Reviewer 9vyj (Part 2)**
> >
> > ***\* The Pile dataset is full of duplicate documents, and this paper does not perform additional deduplication processing.\****
> >
> > **A**: Thank you for pointing out the duplication issue in PILE. We agree that improvements to the dataset, such as deduplication, will help improve the model and training efficiency. We did not do deduplication to PILE because (1) this was only identified recently, so we were not aware of this duplication issue when we were conducting the study, (2) as far as we know, PILE allows the community to train GPT-3 like models, such as GPT-NeoX-20B [7]. MT-NLG 530B [8], etc., and obtain competitive accuracy to GPT-3, so we thought it would be a good data point to add for MoE models (we are not aware of other MoE models trained on PILE), and (3) To our knowledge, PILE is open-sourced dataset, so using it helps the community to reproduce results. Processing PILE would likely lead to different results, but we note that in the worst-case scenario (zero redundancy in MoE), the expert pool does not lead to more parameters than the baseline MoE models, so the exact compression ratio may vary, but that does not invalidate our approach.
> >
> > ***\*  Comparison with dense models. \****
> >
> > **A**: We focused on comparing with MoEs and did not include many comparisons with dense models because there are several prior studies that have already conducted extensive evaluation between MoE and dense model, such as SwithcTransformer[1], GLaM[2], DeepSpeed-MoE[3], GShard[4], Unified Scaling Law [5], we did not feel a need to include the comparison with the dense model given the space limitations. If the reviewer feels there is a strong need to add the results of the dense model, we can incorporate the results in the final version.
> >
> > ***\*  I notice usually a dense model (350M) could get a score of 70.2 on piqa. This SaMoE with more flops achieves a score 68.9. \****
> >
> > **A**: The authors are not sure about which 350M dense model the reviewer refers to. Is it also trained on PILE? If it is trained on a different dataset, it is common to observe slightly different accuracy. Furthermore, a single task indeed has variations. That's why we focused on evaluating multiple tasks (22 tasks) and the average accuracy.
> >
> > [1] Rajbhandari et al. "DeepSpeed-MoE: Advancing Mixture-of-Experts Inference and Training to Power Next-Generation AI Scale.", ICML 2022
> >
> > [2] Du et al. "GLaM: Efficient Scaling of Language Models with Mixture-of-Experts", ICML 2022
> >
> > [3] Fedus et al. "Switch Transformers: Scaling to Trillion Parameter Models with Simple and Efficient Sparsity", https://arxiv.org/abs/2101.03961
> >
> > [4] Lepikhin et al. "GShard: Scaling Giant Models with Conditional Computation and Automatic Sharding", https://arxiv.org/abs/2006.16668
> >
> > [5] Clark et al. "Unified Scaling Laws for Routed Language Models",
> > https://arxiv.org/abs/2202.01169
> >
> > [6] Artetxe et al. "Efficient Large Scale Language Modeling with Mixtures of Experts", https://arxiv.org/pdf/2112.10684.pdf
> >
> > [7] Black et al. "GPT-NeoX-20B: An Open-Source Autoregressive Language Model", https://arxiv.org/abs/2204.06745
> >
> > [8] Smith et al. "Using DeepSpeed and Megatron to Train Megatron-Turing NLG 530B, A Large-Scale Generative Language Model", https://arxiv.org/abs/2201.11990
> >
> > [9] Brown et al. “Language Models are Few-Shot Learners”,  https://arxiv.org/abs/2005.14165

---

> > > ### Comment · Reviewer_9vyj · 2022-11-28
> > > **Thank you for the clarification**
> > >
> > > Thank you for the clarification. I'll leave my rating as it is since it reflects my best judgment of the paper.

---

### Official Review · Reviewer_WUCK · 2022-10-25

**Confidence:** 4
**Correctness:** 3
**Technical Novelty And Significance:** 2
**Empirical Novelty And Significance:** 2
**Recommendation:** 3

**Clarity, Quality, Novelty And Reproducibility:**

The paper is clearly written.
The proposed method is a minor improvement over traditional parameter sharing scheme like suggested in Universal Transformer.
The paper can be reproduced but the results are not valid.

**Strength And Weaknesses:**

Strengths:
- The proposed method is simple in design and implementation but achieves reasonably good results.

Weaknesses:
- The paper does not provide a fair comparison by fixing total parameters but ignore the computational cost (FLOPs) or activated parameters. In traditional MoE research, the general goal is to achieve better quality with a fixed computational cost (FLOPs), not with a fixed total parameters. The reviewer understand that this method provides a efficient way saving total parameters, but the reviewer suspects to achieve better quality, this method would also significantly increase the FLOPs compared to traditional top1 or top2 based routing used in switch transformer and GLaM. Table 1 does not provide any details on computational cost. According to multiple prior works including T5 and the Chinchilla [3], there is always a tradeoff between model capacity and training tokens, the larger the model is, the lower training data/steps can be achieved within the same computational cost budget.  According to Table 1, activated parameters are made fixed around 1B, however, it might not be clear about computational cost in this work's setting.

- The paper misses fair comparisons with many significant related work including autoregressive sparse MoE, GLaM [1]. GLaM adopts a top-2 based routing, that can yield much better results than top-1 based routing. Various efficient routing functions should be compared with in this work, as intelligent routing functions achieve similar effects of improving parameter efficiency. For example, Expert Choice [2] routing achieves heterogeneous experts such that different tokens can utilize a variable number of parameters.

[1] https://arxiv.org/pdf/2112.06905.pdf
[2] https://arxiv.org/abs/2202.09368
[3] https://arxiv.org/abs/2203.15556


**Summary Of The Paper:**

MoEs have been reported to be parameter inefficient such that larger models do not always lead to better performance. This work proposes a parameter-efficient MoE models, by learning a soft combination of a global set of expert layers for each MoE layer. Experimental results show that SaMoE improves parameter efficiency by reducing up to 5.2x parameters while obtaining strong pretraining and zeroshot generalization results.

**Summary Of The Review:**

"No free lunch" in deep learning: reducing parameters and reducing training time will not come for free without sacrificing model quality.
The paper's results are based on fixing total parameters but not on fixing computational cost (activated parameters and FLOPs), which can be unfair to many related works including GLaM. The reviewer would not believe in any results that purely relying on parameter sharing, we could improve quality without introducing additional computational cost. For example, whether this method increases activated parameters (experts per token) is unclear and should be explained. For example, the paper can be increasing the number of layers or expert width or number of experts per token compared to GLaM. All these increase activated parameters, thus inference time. The paper should be also more proactive in explaining why quality gains can be achieved.

---

> ### Author Response · Authors · 2022-11-16
> **Response to Reviewer WUCK  (Part 1)**
>
> ***\*  "No free lunch": The paper does not provide a fair comparison by fixing total parameters but ignore the computational cost (FLOPs) or activated parameters.". \****
>
> **A**: We believe there is some misunderstanding about the objective of this paper. The purpose of many recent MoE works, such as GLaM is to improve **training efficiency**, while our method targets **parameter efficiency**, i.e., how to make better use of the expert parameters.
>
> As stated in our paper (the last part of the second paragraph in the introduction), the main concern of our work was to improve the parameter efficiency of MoE such that the number of trainable parameters of a model is reduced while still arriving at a model with similar or high quality. Although this setting might look different from recently published MoE work, the reviewer pointed out that targeting reducing training cost (e.g., GLaM[6]) or resolving an unbalanced load of experts (e.g., Expert Choice[7]), optimizing for parameter efficiency is motivated by practical constraints. For instance, with expert sharing strategies like SaMoE, it is possible to train the 1.9B MoE model with 8 Nvidia-A100-40G GPUs, whereas the baseline MoE would run out of memory and requires 16 A100 to train the model. Furthermore, once the model has been trained, SaMoE also requires fewer GPUs/TPUs to host the model for inference. Under this setup, model size is more crucial because it raises an issue of feasibility – can we even train or inference a model at all. Finally, we follow the same evaluation setting as prior work on optimization for improved parameter efficiency of DNN models, where the number of parameters is the primary optimization objective, e.g., [1, 2, 3, 4, 5]. In this context, our paper shows that, despite significant prior work on increasing the MoE model scale by increasing parameters, there is significant room for improvement such that MoEs with a smaller number of parameters are able to generalize on par with larger MoEs. We believe the observed parameter efficiency improvement in MoE will motivate subsequent research.
>
> ***\* The paper misses fair comparisons with many related work including autoregressive sparse MoE, GLaM. \****
>
> **A**: We appreciate the reviewer for pointing out GLaM, which we cited and discussed in related work. However, the **GLaM code and dataset are not publicly available**, so we cannot make a direct comparison. Furthermore, we indeed compared SaMoE with two other state-of-the-art open-sourced MoEs from DeepSpeed-MoE [5], which include an autoregressive sparse MoE (denoted as AR-MoE) and a parameter-efficient MoE (denoted as PR-MoE) trained on the publicly accessible dataset PILE.
>
> ***\*  Comparison with various routing functions, such as Expert Choice [7] routing.  \****
>
> **A**: We answer this by stating that SaMoE is oblivious to the choice of routing function. Existing works that improve routing policies also do not exploit expert sharing to improve the parameter efficiency of MoE models like SaMoE. Therefore, expert sharing and routing are orthogonal ideas. To cover the evaluation of SaMoE comprehensively, we choose top-1 gating intentionally to avoid the influence from the routing function side. That said, we are optimistic about the potential benefits of integrating SaMoE into more advanced routing-based solutions to improve their parameter efficiency further. However, we also believe that expert sharing specific routing techniques are required to maximize SaMoE's performance.
>
> ***\* The paper misses fair comparisons with many significant related work including autoregressive sparse MoE, GLaM. GLaM adopts a top-2 based routing, that can yield much better results than top-1 based routing. \****
>
> **A**: To our best knowledge, GLaM did not show (or at least not directly show) comparison results between top-1 and top2. The only explanation of why GLaM uses top-2 gating is in the footnote of Section 4 "setting the number of selected experts to be two is based on the trade-off between predictive performance and the training/serving efficiency of the model." It is unclear how this "trade-off" is measured nor how much better top-2 gating is than top-1. On the other hand, multiple studies such as SwinTransformer[8] and Unified Scaling Law[9] showed through extensive evaluation results that top-1 gating is sufficient to preserve model quality, whereas top-2 routing incurs significant computation and communication overhead than top-1. This is why we focus our studies using top-1 gating in our experiments.

---

> > ### Author Response · Authors · 2022-11-16
> > **Response to Reviewer WUCK (Part 2)**
> >
> > ***\* A minor improvement over traditional parameter sharing scheme like suggested in Universal Transformer. \****
> >
> > **A**: There are multiple differences between our work and Universal Transformers: (1) Universal Transformer manually ties weights of transformer layers, while SaMoE learns a combination of experts, which is completely novel. Our evaluation also shows the effectiveness of SaMoE over manually designed sharing schemes. (2) UT focuses on small-scale dense Transformers, while SaMoE is **the first demonstration of effective expert sharing in multi-billion scale sparse MoE models**.
> >
> > ***\*	Whether this method increases activated parameters (experts per token) is unclear and should be explained. \****
> >
> > **A**: In SaMoE, the experts can be shared across multiple MoE layers. Therefore, everything else being equal (e.g., the same routing function, the same number of MoE layers and experts per layer), the uniquely activated parameters actually decrease compared to the standard MoE, but the number of times a parameter is activated can increase.
> >
> > [1] Mostafa and Wang, "Parameter Efficient Training of Deep Convolutional Neural Networks by Dynamic Sparse Reparameterization", ICML 2019
> >
> > [2] Lan et al. "ALBERT: A Lite BERT for Self-supervised Learning of Language Representations.", ICLR 2020.
> >
> > [3] Takase et al. “Lessons on Parameter Sharing across Layers in Transformers”, ACL 2021
> >
> > [4] Reid et al. "Subformer: Exploring Weight Sharing for Parameter Efficiency in Generative Transformers", EMNLP 2021
> >
> > [5] Rajbhandari et al. "DeepSpeed-MoE: Advancing Mixture-of-Experts Inference and Training to Power Next-Generation AI Scale.", ICML 2022
> >
> > [6] Du et al. "GLaM: Efficient Scaling of Language Models with Mixture-of-Experts", ICML 2022
> >
> > [7] Zhou et al. "Mixture-of-Experts with Expert Choice Routing", https://arxiv.org/abs/2202.09368
> >
> > [8] Fedus et al. "Switch Transformers: Scaling to Trillion Parameter Models with Simple and Efficient Sparsity", https://arxiv.org/abs/2101.03961
> >
> > [9]  Clark et al. "Unified Scaling Laws for Routed Language Models",
> > https://arxiv.org/abs/2202.01169

---

> ### Comment · Reviewer_WUCK · 2022-11-23
> **Thank the authors for their response and clarification.**
>
> It is interesting to provide a parameter efficient view of MoE and this is different from the general trend of MoE research. However, as indicated in many prior work, model parameters determine the modeling power and downstream task complexity the model can handle [1][2][3]. Resonating with Rev 9VYJ, the paper does not train on meaningful pretraining dataset and evaluate on challenging downstream tasks, including translation or question answering.
>
> [1] T5 paper: https://arxiv.org/abs/1910.10683
> [2] Power Law in deep learning: https://arxiv.org/abs/1712.00409
> [3] PaLM paper: https://arxiv.org/abs/2204.02311

---

### Official Review · Reviewer_VcJV · 2022-10-31

**Confidence:** 5
**Correctness:** 3
**Technical Novelty And Significance:** 2
**Empirical Novelty And Significance:** 3
**Recommendation:** 6

**Clarity, Quality, Novelty And Reproducibility:**

The flow of this paper is crystal clear. The authors first identified the bottlenecks of an existing algorithm, found the root cause, proposed a solution, and finally demonstrated the effectiveness of the proposed solution with well prepared experimental evaluations. The text, tables, and figures are all of high quality.


**Strength And Weaknesses:**

Strengths:
1. I think the authors found the right critical bottlenecks for the MoE models, which are the trainability (model quality aspect) and the total number of parameter count (system performance aspect). The proposed solution with the empirical results shows it's on the promising direction,  although it's still far from completely addressing those foundational limitations of MoE,

2. The ablation and scaling laws section are very helpful to the research community to understand how to set the hyperparameters.

Weaknesses:
1. How the speedup in table 1 is evaluated?

2. The gains in the downstream tasks are marginal. It's better to report the variation of zero-shot results at nearby checkpoints as well.

3.  I feel it's important to report the inference step time during autoregressive decoding to best demonstrate the gains from a smaller number of parameters. Because during decoding on accelerators, it's more often memory bound instead of compute bound. When decoding a single token given the prefix, the flops to compute each token is relatively small. However, the whole model parameters (in billions or even trillions) needs to send more HBM to the actual compute units during each decoded step. This HBM-cache communication is usually the dominant factor in the inference cost.

4. Double check the multi-rc results? It's a big jump from 1+ to 70+.


**Summary Of The Paper:**

SaMoE is a novel routing algorithm for mixture of experts (MoE) that allows different MoE layers select experts from a global shared pool. Well designed experiments and scaling law studies are reported in the evaluation section.


**Summary Of The Review:**

This paper worked on an important research topic: how to reduce the training and serving costs for large language models. The proposed algorithm is only marginally novel but empirically significant. The 5x reduction in the number of total parameters would improve the serving speed for MoE models by a lot.

However, the gains in the downstream tasks are only marginal. So it would be better if the authors clearly demonstrated why the large reduction in the parameter count matters using the metrics people care about: the serving latency, the step time, etc. Alternatively, the authors can show the quality difference when matching the inference cost.

---

> ### Author Response · Authors · 2022-11-16
> **Response to Reviewer VcJV**
>
> Thank you for your comments.
>
> ***\*  How the speedup in table 1 is evaluated? \****
>
> **A**: We measured the speedup as the training time of SaMoE over the baseline method (AR-MoE) under the same number of GPUs and the same number of training tokens. We will clarify this in the revision.
>
> ***\* The gains in the downstream tasks are marginal. It's better to report the variation of zero-shot results at nearby checkpoints as well. \****
>
> **A**: We agree. Our initial results show that under the same random seed, nearby checkpoints have very similar zero-shot evaluation results. We will add the full results in the final version.
>
> ***\*  Report the inference step time. \****
>
> **A**: While the main concern of our work was to improve the parameter efficiency of MoE models, for example, we see a 2.6—5.2x total parameter reduction for MoE, we recognize that it would strengthen the work if we also report inference time. Meanwhile, we also recognize that there are few works on how to make effective inferences of MoE models given the challenges the reviewer pointed out: (1) the whole model can be very large that a single device is not sufficient to host the entire model weights, (2) computation and data movement are sparse, so the system optimizations for dense models may need to be adapted or redesigned, and (3) the decoding process makes the computation completely memory-bandwidth bounded. Therefore, the actual saving depends on the hardware and implementation and requires some expansion. We are working on the inference optimizations and will provide some optimized inference results in the revision.
>
> ***\* Double check the multi-rc results? It's a big jump from 1+ to 70+. \****
>
> **A**: Thank you for pointing this out. When generating table 6, the order of the benchmark is slightly misaligned. We did not notice this during submission. We will fix the issue in the revision.

---

> ### Comment · Area_Chair_WWQs · 2022-12-13
> **Notice**
>
> Dear Reviewer,
>
> If you do not want to update your score, please at least acknowledge that you have already read the rebuttal.

---

### Comment · Area_Chair_WWQs · 2022-11-22
**Please respond as soon as possible if you still have questions on the paper.**

Please respond as soon as possible if you still have questions on the paper.

---

### Comment · Area_Chair_WWQs · 2022-11-22
**Please respond as soon as possible if you still have questions on the paper.**

Please respond as soon as possible if you still have questions on the paper.

---

> ### Comment · Area_Chair_WWQs · 2022-11-29
> **Please respond to the authors by Nov. 30**
>
> Please indicate whether the authors' rebuttal addresses your concerns.
>
> If you still have questions, please ask as soon as possible.

---

> > ### Comment · Area_Chair_WWQs · 2022-12-05
> > **Zoom Meeting**
> >
> > For all reviewers, which have not responded to the authors, I will have to ask you to meet via Zoom. If you want to avoid such an additional step, please respond by Dec. 5.

---

### Decision · Program_Chairs · 2023-01-20

**Decision:**

Reject

**Justification For Why Not Higher Score:**

NA

**Justification For Why Not Lower Score:**

NA

**Metareview: Summary, Strengths And Weaknesses:**

This paper presents a parameter-efficient version of Mixture of Experts (MoE), which is known for requiring a large number of parameters to perform well. The proposed model, called SaMoE, uses a soft combination of a global set of expert layers for each MoE layer, which reduces the number of required parameters by up to 5.2 times. The experimental results show that SaMoE not only is more efficient in terms of parameters, but also performs well in pretraining and zero-shot generalization tasks.

The reviewers raised concerns on the evaluation metrics (such as serving latency and step time), comparison fairness regrading baselines, which remain after the discussion. The paper could be significantly strengthened if these concerns can be addressed.

**Summary Of Ac-Reviewer Meeting:**

NA